# CSA: Data-efficient Mapping of Unimodal Features to Multimodal Features

**Po-han Li, Sandeep Chinchali & Ufuk Topcu**
The University of Texas at Austin
{pohanli,sandeepc,utopcu}@utexas.edu

## Abstract

Multimodal encoders like CLIP excel in tasks such as zero-shot image classification and cross-modal retrieval. However, they require excessive training data. We propose canonical similarity analysis (CSA), which uses two unimodal encoders to replicate multimodal encoders using limited data. CSA maps unimodal features into a multimodal space, using a new similarity score to retain only the multimodal information. CSA only involves the inference of unimodal encoders and a cubic-complexity matrix decomposition, eliminating the need for extensive GPU-based model training. Experiments show that CSA outperforms CLIP while requiring $50,000\times$ fewer multimodal data pairs to bridge the modalities given pre-trained unimodal encoders on ImageNet classification and misinformative news caption detection. CSA surpasses the state-of-the-art method to map unimodal features to multimodal features. We also demonstrate the ability of CSA with modalities beyond image and text, paving the way for future modality pairs with limited paired multimodal data but abundant unpaired unimodal data, such as LiDAR and text. [1]

## 1 Introduction

Multimodal encoders like CLIP (Radford et al., 2021) excel in various zero-shot multimodal tasks, such as image classification and cross-modal retrieval. Despite the huge success, they demand excessive training data. For example, OpenAI trained the original CLIP model on 400 million image-text pairs using 592 V100 GPUs, and the training size of the new CLIP models (Ilharco et al., 2021) has increased to 12 billion image-text pairs since then. In addition, more data do not guarantee better performance. CLIP relies on Internet data, which are often of poor quality (Sharma et al., 2018). Incorrectly captioned data may lead to specific failure modes in similar instances (Northcutt et al., 2021; Vasudevan et al., 2022). In this work, we focus on learning a multimodal encoder with limited data that is robust to noisy data.

Our conjecture is that unimodal encoders such as DINO (Oquab et al., 2023) and GTR (Ni et al., 2022) can serve as the building blocks of multimodal encoders. Unimodal encoders only need unimodal data, which are easier to obtain than paired multimodal data. They also require significantly less data than multimodal models, and advanced unimodal encoders are already well-developed. We distill the unimodal knowledge in these models to accelerate the learning of multimodal encoders. We map their unimodal features into a multimodal feature space, similar to how CLIP encodes images and text jointly to a shared feature space. In addition, this process should be data-efficient.

We propose canonical similarity analysis (CSA) to verify our conjecture. It replicates a multimodal encoder such as CLIP with two unimodal encoders, as shown in Figure 1. CSA maps features from unimodal encoders to a multimodal space. It removes redundant information from unimodal features and employs a novel similarity function to replicate CLIP similarity score for multimodal data, *e.g.* images and captions. CSA operates by inference of unimodal encoders and solving an optimization problem, essentially a matrix decomposition without training of neural networks. CSA supports all zero-shot tasks that CLIP is capable of while requiring significantly fewer training data.

---

[1]Project Page: https://d31003.github.io/CSA-Project-Website/

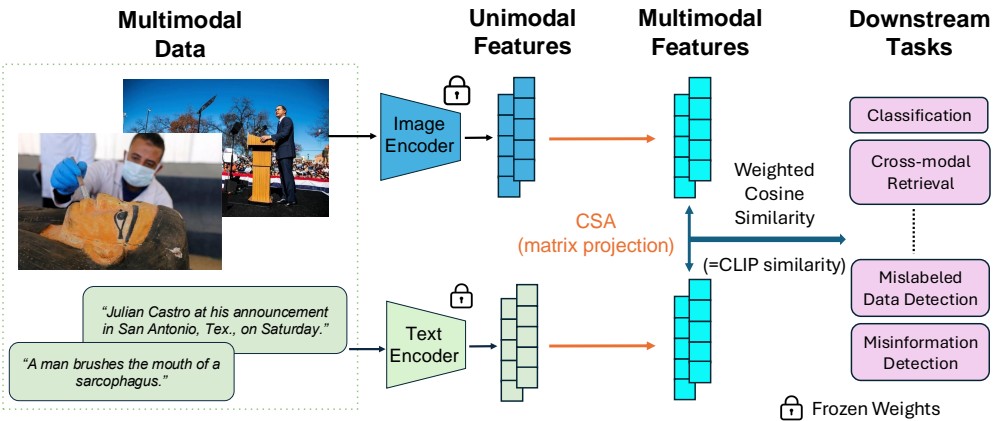

Figure 1: **Canonical similarity analysis (CSA)** replicates the CLIP multimodal similarity scores with two unimodal encoders. CSA uses two unimodal encoders to encode data into unimodal features. Then, it projects the unimodal features to a joint multimodal feature space. The weighted cosine similarity in this feature space replicates the CLIP similarity, enabling various downstream tasks, *e.g.*, cross-modal retrieval. CSA is a robust and data-efficient method that learns even when data are misaligned.

**Contributions.** Our contributions are threefold: (1) We propose CSA which replicates the CLIP model using two unimodal encoders while demanding less computation and data (Section 4). (2) Our theoretical analysis characterizes the trade-off of obtaining informative embeddings and distinguishing multimodal data pairs, considering various hyperparameters of CSA (Section 5). (3) We tested CSA on tasks such as image classification, cross-modal retrieval, and misinformative captions detection (Section 6). The experimental results show that CSA outperforms or matches CLIP while requiring $50,000\times$ fewer paired multimodal data that are used to train the multimodal mapping function given pre-trained unimodal encoders (see Table 1). CSA outperforms the state-of-the-art method that employs unimodal models with the same amount of training data as $18\%$ in ImageNet classification and $23\%$ in image-to-text retrieval.

CSA works across all modalities beyond images and text. We additionally demonstrate its capability with audio and text, paving the way for new modality pairs, such as LiDAR and text Yang et al. (2023). In the future, we envision that new modality pairs will have limited paired multimodal data but sufficient unimodal data, where CSA can efficiently map unimodal features to multimodal features.

## 2 RELATED WORKS

**Unimodal Encoders.** Unimodal encoders encode unimodal data to fixed-dimensional features. In computer vision, encoders, *e.g.*, CosPlace (Berton et al., 2022), ViT (Dosovitskiy et al., 2021), and Dino (Caron et al., 2021; Oquab et al., 2023), are trained with augmented data with random cropping and blurring with contrastive loss (van den Oord et al., 2019). In natural language processing, one common data augmentation method is to randomly mask words or tokens, used in the universal sentence encoder (Cer et al., 2018), Bert (Devlin et al., 2019), and GTR (Ni et al., 2022).

**Multimodal Encoders.** Multimodal encoders project multimodal data points into a feature space shared between modalities. Thus, multimodal features enable training for downstream models (Shridhar et al., 2021; Li et al., 2024b; Goel & Narasimhan, 2024). Another common usage is to directly take the inner product of features of data from different modalities, *e.g.*, an image and a sentence, as a similarity metric. This metric enables downstream tasks such as zero-shot image classification and cross-modal data retrieval. The CLIP model (Radford et al., 2021) is the milestone of multimodal encoders, which bridges the gap between language and vision. CLIP inspires encoders for other modalities, such as audio and inertial measurement units. Such encoders include ImageBind (Girdhar et al., 2023), AudioCLIP (Guzhov et al., 2022) and CLAP (Wu* et al., 2023). Previous work also studied cross-modal retrieval on multi-label images (Ranjan et al., 2015), while ours focuses on a more generic multimodal setting with pre-trained unimodal encoders. Although ImageBind encodes 6 modalities, it lacks training data for all possible modality pairs, resulting in suboptimal performance for pairs such as audio and inertial measurement units.

**Neural Networks with CCA.** Previous works have shown that training losses inspired by canonical correlation analysis (CCA) (Hardoon et al., 2004) are useful in multimodal or multi-distribution tasks (Sun et al., 2019; Lyu & Fu, 2020; 2023). Although taking a crucial step of introducing CCA to deep learning, these works focus on training deep neural networks from scratch instead of using existing foundation encoder models, as in this work. Other works use techniques similar to CCA, such as principal component analysis (PCA), to process the output features of encoders (Li et al., 2024a; Omama et al., 2024). In this work, we focus on using CCA to map unimodal features to a multimodal feature space without any model training, which significantly reduces the size of the required training dataset.

**Combining multiple unimodal Encoders.** The closest work to ours is ASIF (Norelli et al., 2023), which also uses multiple unimodal encoders to bridge the multimodal gap. ASIF calculates the similarity of a pair of multimodal data based on the unimodal similarity of the given data pair and a pre-selected, multimodal anchor dataset, similar to Moschella et al. (2023). It creates a linear kernel space by the anchor set. However, ASIF demands a huge anchor set to achieve similar performance to CLIP, while our proposed method, CSA, constantly outperforms ASIF in all tasks.

## 3  PROBLEM FORMULATION

Given two datasets of size $N$, $X^1 = \{x_1^1, x_2^1, ..., x_N^1\}$, $X^2 = \{x_1^2, x_2^2, ..., x_N^2\}$, from two modalities $m^1, m^2$ (such as vision and language), we want to find two feature extractors (encoders) $\mathbf{E}^1, \mathbf{E}^2$ that map the data to the same-dimensional feature space. The encoders generated feature pairs $\mathbf{E}^1(x_i^1), \mathbf{E}^2(x_i^2)$ are close, and the feature pairs from different pairs of data $\mathbf{E}^1(x_i^1), \mathbf{E}^2(x_j^2)$ are far.

Note that the two sets of data are paired, which means that they are sophisticatedly related (such as an image and its corresponding caption), although of different modalities. Previous works trained multimodal encoder pairs to achieve this goal by minimizing the following loss:

$$\mathcal{L}(X^1, X^2; \mathbf{E}^1, \mathbf{E}^2) := \qquad\qquad\qquad\qquad \text{(Multimodal Contrastive Loss)}$$

$$\frac{-1}{2N}\sum_{i=1}^{N}\left[2(\mathbf{E}^1(x_i^1))^\top(\mathbf{E}^2(x_i^2))) - \sum_{j\neq i}\frac{(\mathbf{E}^1(x_i^1) + \mathbf{E}^2(x_i^2))^\top(\mathbf{E}^1(x_j^1) + \mathbf{E}^2(x_j^2))}{2N-2}\right],$$
$$\tag{1}$$

where the first term encourages similar features for the same pair of data, and the second term encourages the opposite. Variants of this loss are commonly used in contrastive learning (van den Oord et al., 2019), such as CLIP (Radford et al., 2021).

**From Unimodal to Multimodal.** Training a multimodal encoder to learn a multimodal feature space is time-consuming and data-inefficient. The original CLIP model learns from Internet-scale datasets (400 million images) and requires 500 A100 Nvidia GPUs. Instead of training end-to-end multimodal encoders, we leverage existing unimodal encoders to accelerate the multimodal learning process. That is, given unimodal encoders that encode data into unimodal feature spaces, we directly find two mapping functions that map each unimodal feature space to a multimodal space.

Mathematically, we use two unimodal encoders $\mathbf{E}^1, \mathbf{E}^2$ that capture only unimodal latent information and individually encode data $x_i^1, x_i^2$ into features $\hat{z}_i^1, \hat{z}_i^2 \in \mathbb{R}^{q^1}, \mathbb{R}^{q^2}$. Because these features $\hat{z}_i^1, \hat{z}_i^2$ come from different encoders, their dimensions need not be the same ($q^1 \neq q^2$), which complicates the discovery of the mapping functions. The dimensions of the features are significantly smaller than those of the raw data $x_i$. Hence, we utilize unimodal encoders to significantly reduce the dimension of the data and the complexity of the problem. Our ultimate goal is to efficiently find two mapping functions $\mathbf{A}, \mathbf{B}$ that map the feature spaces to a multimodal one while preserving the contrastiveness property characterized in Equation 1.

## 4  CANONICAL SIMILARITY ANALYSIS (CSA)

The problem in Section 3 is easier to state than to solve. First, unimodal encoders have completely different architectures (*e.g.*, transformers vs. U-net), losses, and output dimensions. Second, mapping data from various-dimensional feature spaces to a multimodal one means that we need to discard latent information from the higher-dimensional space. The challenge lies in determining what

information to discard to maintain the contrastiveness property. Also, we do not want to preserve all information as there might be some modality-specific information that we wish not to preserve through the mapping, such as words that cannot be expressed in images or punctuation.

We thus propose canonical similarity analysis (CSA). We observe that all downstream tasks use cosine similarity to obtain the similarity of multimodal data points. It inspires us to use correlation coefficients, the cosine similarity between centered vectors, to approximate this similarity and build the mapping functions from unimodality to multimodality. Precisely, we find the pairs of bases in each unimodal feature space that maximize the correlation coefficients. We use CCA to find such bases. Then, instead of directly using an inner product, we use a weighted cosine similarity score to evaluate the similarity of the multimodal data. We discard information from low-correlated bases, as they might not be relevant to the multimodal feature space, and we weight the contributions of the bases to the similarity score based on their correlation coefficients.

## 4.1 CANONICAL CORRELATION ANALYSIS

Recall the previous notation of two sets of data $X^1 = \{x_1^1, x_2^1, ..., x_N^1\}$, $X^2 = \{x_1^2, x_2^2, ..., x_N^2\}$. We use two unimodal encoders $\mathbf{E}^1, \mathbf{E}^2$ to encode and normalize the data into zero-meaned latent features $\{\hat{z}_i^1, \hat{z}_i^2\}_{i=1}^N$. To find the bases that maximize the correlation coefficients, we use CCA to map the two feature spaces to a multimodal feature space of dimension $r = \min(q^1, q^2)$:

$$\mathbf{A}^*, \mathbf{B}^* = \underset{\mathbf{A} \in \mathbb{R}^{r \times q^1}, \ \mathbf{B} \in \mathbb{R}^{r \times q^2}}{\arg\max} \quad \mathrm{Tr}\left(\mathbf{A}\hat{Z}^1\left(\mathbf{B}\hat{Z}^2\right)^\top\right) \qquad \text{(Formulation of CCA)}$$

$$\text{s.t.} \quad (\mathbf{A}\hat{Z}^1)(\mathbf{A}\hat{Z}^1)^\top = (\mathbf{B}\hat{Z}^2)(\mathbf{B}\hat{Z}^2)^\top = \mathbf{I}_r, \tag{2}$$

where $\hat{Z}^1 \in \mathbb{R}^{q^1 \times N}, \hat{Z}^2 \in \mathbb{R}^{q^2 \times N}$ are the matrices consisting of column feature vectors $\hat{z}$, $\mathbf{I}_r \in \mathbb{R}^{r \times r}$ is the identity matrix, and $\mathrm{Tr}$ is the trace function. The objective of Equation 2 can be viewed as the sum of the correlation coefficients of the bases, and the identity constraints normalize the vectors so that covariance becomes correlation. In summary, CCA finds the bases of the two given matrices such that the projected vectors have the highest correlation coefficients $\rho_i$ for $i = 1, ..., r$.

Solving CCA is similar to principal component analysis (PCA), which is essentially a singular value decomposition (SVD) of time complexity $O(q^1 q^2 r)$. Therefore, it possesses the beneficial property of removing noise from the data, which we later detailed in the theoretical analysis. Weenink (2003) gives the analytical solution:

$$\mathbf{A}^*, \mathbf{B}^* = U\Sigma_{\hat{z}^1}^{-1/2}, V\Sigma_{\hat{z}^2}^{-1/2}, \quad \text{(Analytical Solution of CCA)} \tag{3}$$

where $U^\top PV$ are the resulting SVD matrices of the rank-$r$ matrix $= \Sigma_{\hat{z}^1}^{-1/2}\hat{Z}^1\hat{Z}^{2\top}\Sigma_{\hat{z}^2}^{-1/2} \in \mathbb{R}^{q^1 \times q^2}$. The correlation coefficients $\rho$ of Equation 2 are the diagonal entries of $P$ in descending order $\rho_1 \geq \rho_2 \geq ... \geq \rho_r$ like the descending principal components of PCA.

## 4.2 CANONICAL SIMILARITY ANALYSIS

We now define CSA. For any pair of multimodal data $\{x_j^1, x_j^2\}$, with slight abuse of the notation of original data and features, we define the similarity as:

$$\mathcal{S}(x_j^1, x_j^2; s) = \mathcal{S}(\hat{z}_j^1, \hat{z}_j^2; s) = \frac{\sum_{i=1}^s \rho_i (\mathbf{A}^*\hat{z}_j^1)_i (\mathbf{B}^*\hat{z}_j^2)_i}{\|(\mathbf{A}^*\hat{z}_j^1)_{1:s}\|_2 \|(\mathbf{B}^*\hat{z}_j^2)_{1:s}\|_2}, \qquad \text{(Canonical Similarity)} \tag{4}$$

where $(\cdot)_{1:s}$ denotes the 1st to $s$-th row vector, $(\cdot)_i$ denotes the $i$-th entry, and $s$ is a pre-defined hyperparameter. Note that Equation 4 is the weighted cosine similarity of the first $s$ dimensions of the transformed data matrix. The weights determine how much each dimension influences the overall similarity score, depending on their level of correlation. We chose the hyperparameter $s$ so that CSA preserves the information of dimensions with correlations greater than $\rho_s$. As stated earlier, the feature space might contain modal-specific information that cannot be mapped to the other modality. Thus, we only take into account meaningful bases. We choose CCA with linear correlation instead of complicated kernel CCA because most contrastive learning loss functions only contain the linear inner product of the feature vectors, which is also the case in Equation 2.

### 4.3 TRAINING AND INFERENCE

Similar to standard machine learning models, CSA consists of both training and inference phases. We split the multimodal data into training and test sets, just like in standard machine learning. CSA performs all optimizations on the training set, and the test set is not known a priori and is used only for evaluation. During the training phase, or also called the fitting phase, CSA obtains the training set of encoded features from the unimodal encoders and then solves Equation 2. Note that CSA does not train any encoders, so it does not require GPUs. The resulting correlation coefficients $\rho$ determines the value of $s$ in Equation 4:

$$s = \arg \min_i \rho_i \quad \text{s.t. } \rho_i \geq \text{const.} \tag{5}$$

Note that $\text{const}$ here is an empirical constant threshold to optimize performance in our experience. Finally, for any multimodal downstream task, CSA uses Equation 4 to evaluate the similarity between any pair of multimodal data. We later show the experimental results of CSA and compare the results with the state-of-the-art methods.

## 5 THEORETICAL ANALYSIS OF CSA

In this section, we characterize the trade-off of obtaining informative embeddings and distinguishing multimodal data pairs caused by selecting the appropriate $s$ in Equation 4. We start with the linear setting, but the conclusion holds for a more general setting.

**Assumptions.** We follow the assumption of data generation as per Joshi et al. (2024) and analyze the effect of CSA based on the theoretical results of contrastive losses (Ji et al., 2023; Tian, 2022; Nakada et al., 2023). Suppose that the $N$ pairs of data $\{x_i^1, x_i^2\}_{i=1}^N$ from two modalities $m^1, m^2$ are generated as follows:

$$x_i^1 = \mathcal{T}^1 z_i + \epsilon_i^1, \quad x_i^2 = \mathcal{T}^2 z_i + \epsilon_i^2. \tag{6}$$

Here, $z_i \in \mathbb{R}^q$ is the shared unobservable latent feature of the two modalities, and $\mathcal{T}^1 \in \mathbb{R}^{p^1 \times q}, \mathcal{T}^2 \in \mathbb{R}^{p^2 \times q}$ are the linear mapping functions of the latent vectors to the observable data. Without loss of generality, we assume that both the latent feature vectors and noise for each modality are independent and identically distributed (i.i.d.) samples from any distribution.

**Optimal Linear Encoders of Contrastive Loss.** Previous work (Ji et al., 2023) gives the analytical solution of a unimodal linear encoder $\mathbf{E}^i$ trained on linear contrastive loss with norm regularization:

$$\mathcal{L}(X^k, X'^k; \mathbf{E}^k) := \frac{-1}{2N} \sum_{i=1}^N \left[ 2(\mathbf{E}^k x_i^k)^\top (\mathbf{E}^k x_i'^k) - \sum_{j \neq i} \frac{(\mathbf{E}^k x_i^k + \mathbf{E}^k x_i'^k)^\top (\mathbf{E}^k x_j^k + \mathbf{E}^k x_j'^k)}{2N - 2} \right]$$
$$+ \frac{\lambda}{2} \|\mathbf{E}^k \mathbf{E}^{k\top}\|_F^2, \qquad \text{for } k = 1, 2, \qquad \text{(Unimodal Contrastive Loss)} \tag{7}$$

where $X^k, X'^k$ are two sets of original and augmented (*e.g.*, random masked, Gaussian blurred, etc.) data from the same modality, and $\lambda$ is a hyperparameter weighting the regularization term. Note that this contrastive loss function is similar to Equation 1, but the second modality is the augmented data now. For simplicity, we assume that all unimodal data are augmented with complementary random masking with probability $0.5$, so the optimal linear encoder is:

$$\mathbf{E}_{\text{linear}}^k = \arg \min_{E \in \mathbb{R}^{q \times p^k}} \mathcal{L} = C \left( \sum_{i=1}^q u_i \sigma_i e_i^\top \right)^\top, \quad \text{for } k = 1, 2. \tag{8}$$

In Equation 8, $C > 0$ is a positive constant related to $\lambda$ in Equation 7. $e_i$ is the $i$-th standard vector basis, and $\sigma_i$ is the $i$-th largest eigenvalue of matrix:

$$\text{OffDiag}(X^i X^{i\top}) - \frac{1}{N-1} X^i (\mathbf{1} - \mathbf{I}_N) X^{i\top}, \tag{9}$$

where OffDiag denotes the function that makes all diagonal entries 0, and $\mathbf{1}$ is a square matrix of ones. From Equation 3, 6, and 8, the solution of CSA in linear setting is:

$$\mathbf{A}^* \hat{z}_i^1 = \mathbf{A}^* \mathbf{E}_{\text{linear}}^1 x_i^1 = \mathbf{A}^* \mathbf{E}_{\text{linear}}^1 (\mathcal{T}^1 z_i + \epsilon_i^1), \quad \mathbf{B}^* \hat{z}_i^2 = \mathbf{B}^* \mathbf{E}_{\text{linear}}^2 x_i^2 = \mathbf{B}^* \mathbf{E}_{\text{linear}}^2 (\mathcal{T}^2 z_i + \epsilon_i^2). \tag{10}$$

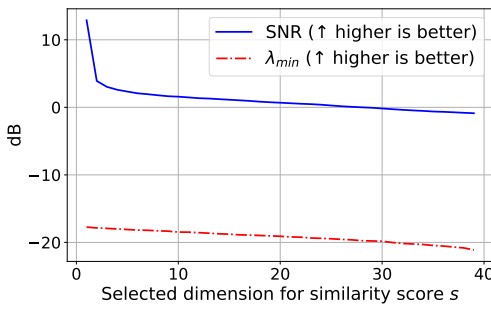 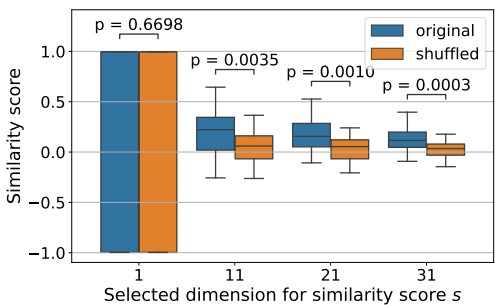

(a) SNR and $\lambda_{\min}$ vs. selected dimension.      (b) p-value (lower is better) of similarity scores.

Figure 2: **Trade-off of canonical similarity analysis:** (a) When $s$ is low, the signal-to-noise ratio (SNR) of the data is high, and the minimum singular value $\lambda_{\min}$ is large. It preserves the distance between contrastive data. (b) When $s$ is low, the p-value is large, and the original and shuffled distributions are alike, so the similarity score is meaningless. (a) shows the desirable properties when $s$ is low and (b) shows the opposite.

**Sensitivity to Noise.** The noise in Equation 6 affects the unimodal linear encoders $\mathbf{E}^1_{\text{linear}}, \mathbf{E}^2_{\text{linear}}$ from the second term of Equation 9. It hinders the unimodal encoders in finding the inverse of $\mathcal{T}^1, \mathcal{T}^2$ but in encoding the noise. The noise then propagates and affects the covariance matrices of $\hat{z}$ and the decomposed orthogonal matrices $U$ and $V$ in Equation 3,. Hence, we rely on the hyperparameter $s$ to prevent noise from dominating. When $s$ increases, we include the less correlated dimensions caused by noise in the canonical similarity metric in Equation 4.

**Distinguish Contrastive Data Pairs.** In Equation 7, similar sample pairs, *i.e.*, the augmented and original data, stay close to each other, while dissimilar ones, *i.e.*, augmented data from different data points, are far apart. However, in the objective of CCA Equation 2, we focus only on maximizing the correlation coefficients of similar samples while ignoring dissimilar ones.

We now analyze how CSA distinguishes contrastive latent feature pairs from the two unimodal encoders. Given two data pairs $(x^1_a, x^2_a)$ and $(x^1_b, x^2_b)$, we assume that the predicted latent feature pairs $(\hat{z}^1_a, \hat{z}^1_b), (\hat{z}^2_a, \hat{z}^2_b)$ are distant because the unimodal encoders are trained for it. The distance between latent features preserved in the matrix transformations is bounded as follows:

$$\lambda_{\min}(\mathbf{A}^*)\lambda_{\min}(\mathbf{E}^1_{\text{linear}})\|x^1_a - x^1_b\|_2 \le \lambda_{\min}(\mathbf{A}^*\mathbf{E}^1_{\text{linear}})\|x^1_a - x^1_b\|_2 \le \|\mathbf{A}^*(\hat{z}^1_a - \hat{z}^1_b)\|_2,$$
$$\lambda_{\min}(\mathbf{B}^*)\lambda_{\min}(\mathbf{E}^2_{\text{linear}})\|x^2_a - x^2_b\|_2 \le \lambda_{\min}(\mathbf{B}^*\mathbf{E}^2_{\text{linear}})\|x^2_a - x^2_b\|_2 \le \|\mathbf{B}^*(\hat{z}^1_a - \hat{z}^1_b)\|_2, \tag{11}$$

where $\lambda_{\min}(\cdot)$ denotes the minimum singular value, which is the opposite of the Lipschitz constant of a matrix. From Equation 11, we can see that the distance between the CSA features (right-hand side) is lower bounded by the distance of the original data with a constant of minimum singular values (left-hand side). Thus, we know how the hyperparameter $s$ selecting the dimension of CSA affects the distance between dissimilar data pairs. When the hyperparameter $s$ decreases, CSA takes into account fewer dimensions per Equation 4. That is, $\lambda_{\min}(\mathbf{A}^*_{1:s}) = \rho_s$ is effectively larger, which ultimately results in more distant features in the CSA space, and vice versa. In summary, a smaller $s$ preserves distance in the CSA space, which distinguishes contrastive data pairs.

**Trade-off of Canonical Similarity Analysis.** Previous results show that a smaller $s$ leads to less noisy but more contrastive similarity scores, which is desirable. However, $s = 1$ is not desirable because it loses much information about the true latent features, and most vectors will have identical similarity scores. In practice, we do not know the dimension of the latent feature $q$ and can only use CSA to project the encoded features of two modalities to a dimension that is less noisy, but also contains enough information about the true shared latent feature $z$.

**Numerical Experiments.** We validate the trade-off through numerical experiments following Equation 6. Figure 2a shows that if we only account for a few dimensions ($s$ is low), the signal-to-noise ratio (SNR) of the data is then high, and the minimum singular value $\lambda_{min}$ is large, which preserves the distance between contrastive data. Note that Figure 2a shows the values in decibels (dB). Figure 2b compares paired data $(x^1_i, x^2_i)$ and randomly shuffled data $(x^1_i, x^2_j), i \ne j$. High Wilcoxon p-values mean that the two distributions of similarity scores from Equation 4 are alike. In this case, the canonical similarity score is meaningless, and larger $s$ is more desirable. The intrinsic trade-off

| Method | Train images | Train text | Parameter | Feature dimension $q, r$ |
|---|---|---|---|---|
| CLIP | 2B | 2B | 1.3B | $q = 1280$ |
| GTR | ✗ | 2B | 335M | $q = 768$ |
| DINOv2 | 142M | ✗ | 1.1B | $q = 1536$ |
| CSA (ours on ImageNet) | 35k | 1k | ✗ | $r = 700$ |
| CSA (ours on Leafy Spurge) | 800 | 2 | ✗ | $r = 250$ |
| CSA (ours on Flickr30k) | 5k | 25k | ✗ | $r = 200$ |
| CSA (ours on COSMOS) | 41k | 41k | ✗ | $r = 750$ |

Table 1: **Comparison of methods:** CSA utilizes two unimodal encoders to match the performance of CLIP while requiring significantly less data. We show the CLIP parameters of only one encoder for a fair comparison to other unimodal encoders.

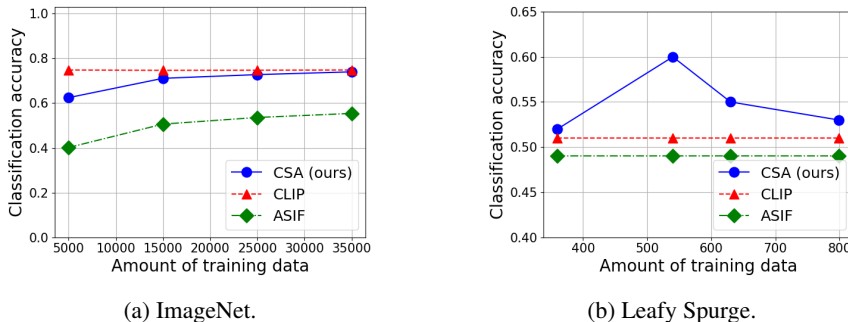

(a) ImageNet.           (b) Leafy Spurge.

Figure 3: **Image classification:** CSA (blue) is highly data-efficient, requiring only $35,000$ training samples to match the performance of CLIP in ImageNet and 360 samples in Leafy Spurge.

| Method | mAP@5 | Precision@1 | Precision@5 |
|---|---|---|---|
| CSA (ours) | 36.6% | 59.3% | 43.4% |
| CLIP | 73.8% | 92.9% | 77.2% |
| ASIF | 14.6% | 25.6% | 20.0% |

| Method | Precision@1 |
|---|---|
| CSA (ours) | 44.7% |
| CLIP | 79.5% |
| ASIF | 0.1% |

(a) Image-to-text retrieval          (b) Text-to-image retrieval

Table 2: **Cross-modal retrieval on Flickr30k:** CSA exhibits lower performance compared to the multimodal baseline, CLIP. However, it outperforms the unimodal baseline, ASIF, in both retrieval scenarios.

in selecting $s$ is now clear. When $s$ is too small, we do not have meaningful similarity scores that distinguish unpaired data, but when $s$ is too large, we have noisy and closer clusters of features.

## 6 EXPERIMENTS

**Baselines.** We compared CSA with another baseline, ASIF (Norelli et al., 2023). It has the same setting as CSA, which uses two unimodal encoders to calculate the similarity of a multimodal data pair. Unlike CSA, which solves an optimization problem, ASIF infers multimodal similarity by the unimodal distance between data points. ASIF is the only fair baseline for CSA. We trained CSA and ASIF with the same training set and unimodal encoders and evaluated all the methods in the test set. We also compared CSA with the state-of-the-art multimodal encoder model from OpenCLIP (Ilharco et al., 2021; Cherti et al., 2022), acknowledging their incomparable training set and model parameters, shown in Table 1. In addition to encoder models, we compared the performance of LLaVA2-13B (Touvron et al., 2023), a multimodal large language model, and prompted it to solve the specified tasks from the provided image and text. The unimodal encoders of CSA and ASIF are GTR (Ni et al., 2022) and DINOv2 (Oquab et al., 2023) here. Section A shows the detailed settings.

**Image Classification.** We first examined CSA's ability of image classification on ImageNet and Leafy Spurge. All methods input an image and all possible captions "This is an image of *label*" and select the most similar caption as the predicted label. In Figure 3a, CSA requires only 35,000 training samples to match the performance of CLIP and consistently outperforms ASIF, which needs millions of data points to achieve a similar performance to CLIP. Since CLIP is zero-shot, its performance is independent of the amount of data. Leafy Spurge challenges the capability of methods with

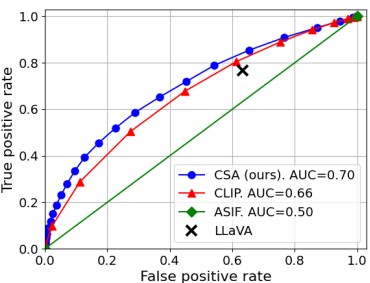
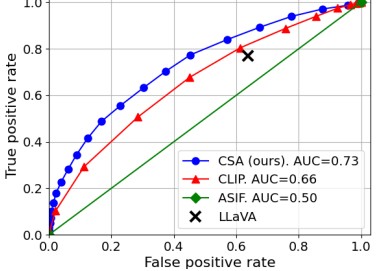

(a) Results for a training size of 5000.       (b) Results for a training size of 35000.

Figure 4: **Detecting mislabeled ImageNet images:** CSA (blue) outperforms CLIP, ASIF, and LLaVA with a higher AUC. (a) and (b) illustrate the results for CSA and ASIF across 2 training set sizes, showing the superior performance of CSA with limited and noisy training data.

extremely limited data. This dataset includes only 800 training images and 100 test images of plants with binary classes: leafy spurge and others. The images are out-of-distribution for any encoders, as this dataset was newly published in 2024 and captured using drones. Figure 3b shows that while the performance of CSA fluctuates, it consistently outperforms CLIP and ASIF in this challenging scenario, demonstrating CSA's effectiveness with limited out-of-distribution data.

Table 1 shows the size of the training set and the number of parameters of the neural network encoders. Notably, CSA surpasses CLIP by utilizing two unimodal model encoders trained on 2 billion text samples and 142 million images, along with only 35,000 multimodal data points. This is $50,000\times$ less unimodal data compared to the CLIP training set. We also visualize the image features of CSA and CLIP in Section H.

**Cross-modal Retrieval.** We examined CSA and the other baselines on a cross-modal retrieval dataset, Flickr30k (Young et al., 2014), which contains 15,000 images, each paired with 5 textual descriptions. This dataset is common for evaluating the performance of image-to-text and text-to-image retrieval. For image-to-text retrieval, the task is to find the corresponding caption from a set of possible candidates (the so-called reference set) given a query image and vice versa. We retrieved the most similar images or text from the reference set according to the similarity scores of all methods and showed the results in Table 2. For image-to-text retrieval, we show mean average precision (mAP), precision@1, and precision@5. For text-to-image retrieval, we only evaluated performance using precision@1 since there is only one correct image. From Table 2, CSA exhibits lower performance compared to the multimodal baseline, CLIP. However, it outperforms the unimodal baseline, ASIF, in both retrieval scenarios. Notably, ASIF completely fails in the text-to-image retrieval task.

**Detecting Mislabeled ImageNet Images.** We again show CSA's superiority with limited and noisy data. Previous works show that ImageNet is imperfect and contains incorrectly labeled images, leading to several failure modes of downstream models (Vasudevan et al., 2022). Trained with the original incorrect ImageNet dataset, all methods detect if the input image and the label align. We use human-evaluated labels from (Northcutt et al., 2021) to define incorrectly labeled images.

All methods input an image and caption "This is an image of *label*" and output if the image and caption align or not (true or false). LLaVA answers questions about whether the given image and caption align. CSA, CLIP, and ASIF output true if the calculated similarity score of the image and caption pair is greater than a threshold. We enumerated the thresholds to obtain several true and false positive rates and show the receiver operating characteristic (ROC) curves in Figure 4. In the legend, we listed the area under the curve (AUC), where a higher value indicates better performance. In Figure 4, CSA outperforms all other baselines, even LLaVA, given 5,000 image-label pairs. We also see that when the size of the training data increases from 5,000 to 35,000, the performance of CSA and ASIF increases as well. We emphasize that the comparison of AUC does not directly compare accuracy. As shown in Figure 4b, the true positive rate of CSA outperforms LLaVA by nearly 10% under the same false positive rate.

Note that the training sets of CSA and ASIF contain mislabeled ImageNet images, which are 11% of the training data in both dataset sizes. It shows CSA 's robustness to noisy data and efficiency in learning from just 5,000 pairs. Similarly, the CLIP encoders' training set also contains mislabeled

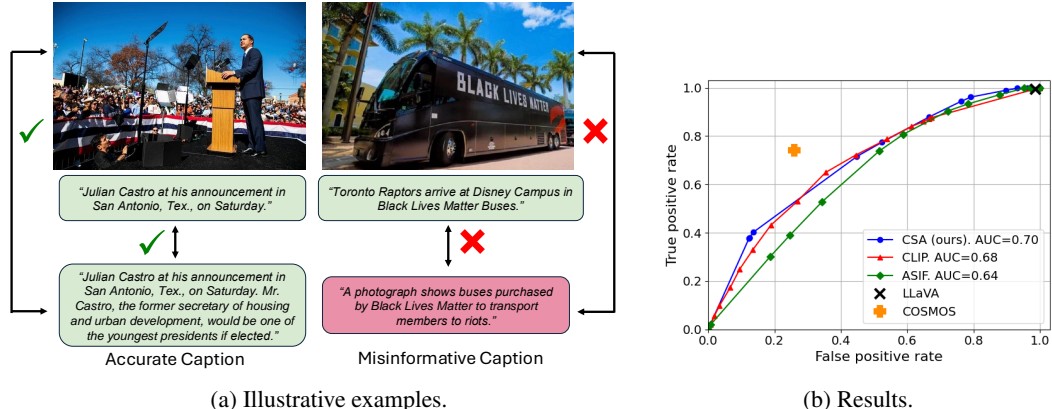

(a) Illustrative examples.

(b) Results.

Figure 5: **Detecting misinformative news captions:** (a) We consider the retrieved captions misinformative if they do not align with the images and the corresponding captions. (b) CSA (blue) outperforms CLIP, ASIF, and LLaVA with a higher AUC. The orange cross is the supervised learning method from the original COSMOS paper trained with labels of object locations, which is the only method that outperforms CSA.

ImageNet images, making the comparison fair. We later demonstrate CSA 's robustness with even noisier training data.

**Detecting Misinformative Captions.** Detecting mislabeled images is easy since text captions differ only in labels. We considered a much harder task to detect misinformation news captions from the COSMOS dataset (Aneja et al., 2023). The task involves determining whether a Google-retrieved caption aligns with news images and its original captions, as shown in Figure 5a. The benchmark considers the retrieved captions misinformative if they do not align with the images and the corresponding captions simultaneously. The ultimate goal of the task is to identify and prevent the spread of misinformative news captions on the Internet.

We evaluated the similarities between the image and its retrieved caption, as well as between the two captions, and identified misinformation when both similarity scores fell below the respective thresholds. Figure 5b shows all the ROC curves and their AUC. Again, CSA outperforms CLIP with two unimodal encoders. The supervised learning method from the original COSMOS paper is the only method that performs better than CSA, although trained with supervised labels of object locations. Note that the LLaVA agent completely fails this task with a false positive rate of $100\%$, as it always thinks that the retrieved captions align with the given images and the original captions.

**Robustness to noisy data.** To test CSA's robustness, we randomly shuffled a percentage of the training labels of ImageNet and evaluated the resulting test accuracy. Figure 6 shows that CSA constantly outperforms ASIF and achieves $70\%$ accuracy even if $50\%$ of the training labels are incorrect. When all the data are shuffled, both CSA and ASIF degrade to $0\%$ accuracy.

**Towards more modalities and ablation study.** We show additional results in the Appendix to provide a comprehensive study to CSA. To show that CSA generalizes to other modalities, we tested it on audio and text data in Section B, text-to-LiDAR retrieval in Section F, and text-to-timeseries in Section G. See Section C for the correlation coefficients of multimodal data and the results of CSA with various $s$ in Equation 4 in Section D and uni-

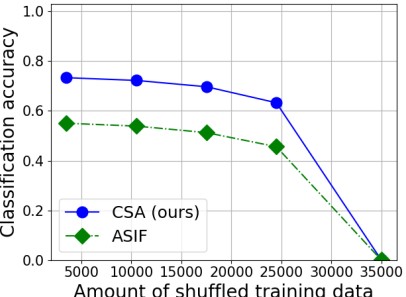

Figure 6: **Robustness to noisy ImageNet data:** CSA remains performant even when the training labels, comprising a total of 35,000 samples, are partially shuffled.

modal encoders in Section E. We visualized and examined the CSA embeddings in Section H and Section I. Lastly, we present the ablation study of CLIP encoder architectures in Section J.

**Limitations.** CSA is suitable for bimodal data, similar to CLIP, unlike Imagebind (Girdhar et al., 2023) is capable of 6 modalities. Like ASIF, CSA is based on unimodal encoders. If the unimodal encoders are not foundation models but trained on limited datasets like MNIST, the performance of CSA degrades. Once the encoded features are obtained, CSA is just SVD. In our case, the underlying

implementation of CSA is NumPy, which is efficient in handling large matrix operations parallelly on multiple CPUs. Also, solving SVD with NumPy takes more time and is computationally efficient than training models on GPUs for datasets of the same size. There are tricks to speed up SVD approximately. For instance, GPU libraries like CuPy Okuta et al. (2017), and randomized SVD algorithms compute approximate results faster than full SVD. Lastly, for a given hyperparameter $s$, we only need to obtain the first $s$ singular values, which can also speed up the computation.

**Summary.** CSA always outperforms the unimodal baseline, ASIF, and matches or exceeds CLIP's performance, except in cross-modal retrieval. This exception likely arises because cross-modal retrieval involves comparing thousands of similarity score pairs between the query data and the reference set, unlike the other tasks that compare against certain captions. It highlights the trade-off of CSA discussed in Section 5. CSA encounters a trade-off between distinguishability and informativeness, *i.e.*, to maintain the distant features. Retrieval tasks require a more curated balance between these aspects, and other tasks benefit from greater distinguishability of similarity scores.

We conclude that CSA is a robust and data-efficient method to replicate the CLIP similarity score using two unimodal encoders. In Table 1, the amount of data required compared to CLIP underscores that unimodal encoders are significantly more efficient to train. A small amount of multimodal data suffices to learn the mapping of unimodal features to a multimodal space, even if the training data are unprocessed by humans (COSMOS) or incorrectly labeled (ImageNet).

# 7 CONCLUSIONS AND FUTURE WORKS

In this work, we proposed CSA, which maps two unimodal feature spaces from pre-trained encoders to a multimodal feature space. CSA is extremely data-efficient. We characterize the intrinsic trade-off of CSA, and CSA shows competitive performance compared to CLIP models in image classification, mislabeled data detection, text-to-LiDAR retrieval, and misinformation detection tasks with limited data. CSA also outperforms the state-of-the-art method in the same setting.

Future work includes extending CSA to more than 2 modality pairs, just as generalized CCA (Horst, 1961) extends CCA to more sources. In addition, understanding the relationship between the size of the training set and its performance is crucial. If we can fine-tune the unimodal encoders, which loss function will result in the most suitable feature space for CSA remains an open problem. Lastly, CSA essentially finds a mapping function between two feature spaces, which need not be modality features. We aim to test CSA on mapping intramodal data, such as multi-view images of an object.

## REPRODUCIBILITY STATEMENT

For the theoretical analysis of this work, we state all assumptions made in the **assumptions** paragraph. For all the hyperparameters and detailed settings of the experiments, please refer to Appendix Section A. Lastly, we put the core code of CSA in the supplementary details. The code includes dataloaders, execution code, and links to download all the datasets and models used. We will provide an example configuration file after publication, as it contains non-anonymous directory paths.

## ACKNOWLEDGEMENT

This work was supported in part by the National Science Foundation grants No. 2133481 and No. 2148186, NASA 80NSSC21M0071, the Office of Naval Research (ONR) under Grant No. N00014-22-1-2254, the Defense Advanced Research Projects Agency (DARPA) contract FA8750-23-C-1018, and DARPA ANSR: RTXCW2231110. Any opinions, findings, and conclusions or recommendations expressed in this material are those of the authors and do not necessarily reflect the views of the National Science Foundation. We also thank Mohammad Omama and Sai Shankar Narasimhan for their valuable feedback and discussions on the paper.

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

# Appendices

## A  ADDITIONAL DETAILS ON THE EXPERIMENTS

We run all inferences of LLAVA and encoder models on an NVIDIA RTX A5000 GPU, and solving Equation 2 with 35,000 multimodal feature vectors on a 64 core Xeon Gold 6226R CPU machine takes less than 10 minutes. The implementation of CCA is from CCA Zoo Chapman & Wang (2021).

**Multimodal Encoders.** The multimodal image-text encoder used throughout the experiments is *laion/CLIP-ViT-bigG-14-laion2B-39B-b160k* from Huggingface. The multimodal audio-text encoder used is *laion/larger_clap_general* from Huggingface (Wu* et al., 2023).

**Unimodal Encoders.** We use several unimodal encoders and show the difference in performance in Section E. To encode images, we tested DINOv2-Giant (Oquab et al., 2023) and the unimodal part of the multimodal encoders previously mentioned. To encode text, we tested GTR-t5-large (Ni et al., 2022) and the unimodal part of the multimodal encoders mentioned above. To show CSA's ability to combine unimodal models, we never tried using the paired unimodal encoders of a multimodal encoder in our experiments, *i.e.*, using CLIP to encode both images and text.

**Flickr30k.** We trained ASIF and CSA on the Flickr validation set, which includes 145,000 images and 5 captions for each image. We then validated the models on a test set of 5,000 images and 25,000 captions.

**COSMOS.** We trained ASIF and CSA on the COSMOS validation set, which includes 41,006 image-caption pairs. We then validated the models using a test set of 1,700 image-caption pairs, with half of the captions labeled as misinformation by human annotators.

## B  TOWARDS MORE MODALITIES—AUDIO AND TEXT

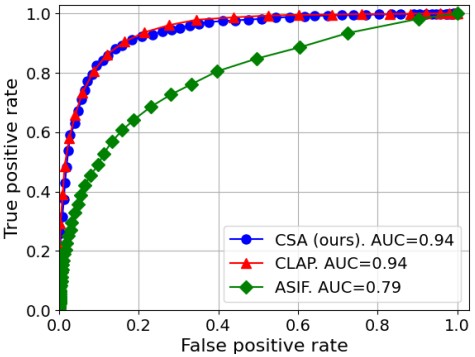

Figure 7: **Classification of YouTube audio and genre tags:** CSA (blue) performs as well as CLAP, the CLIP-inspired multimodal audio and text encoder, and outperforms ASIF in classifying genre tags of YouTube audio.

We now show CSA's generalization ability to more modalities with MusicCaps (Agostinelli et al., 2023). We use GTR and CLAP to encode YouTube audio along with the tagged genre descriptions of the audio. We conducted a classification task in which the models input the audio and a tag and output if the audio aligns with the caption. Similar to the mislabeled ImageNet experiment, we show the ROC curves and compare the AUC in Figure 7. We trained ASIF and CSA for 3,777 data points and tested all methods on 1,625 data points. We randomly sampled a tag for each data point during both training and inference. In Figure 7, we see that CSA performs as well as CLAP, the CLIP-inspired multimodal audio and text encoder, and outperforms ASIF. Thus, we conclude that CSA extends its capabilities beyond image and text, effectively handling audio and text as well.

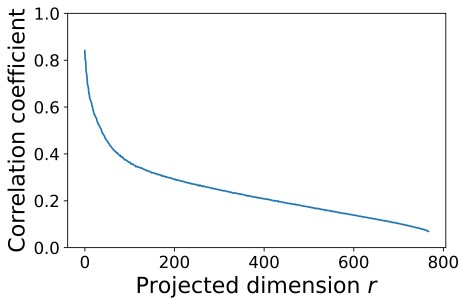

Figure 8: **Correlation coefficients of COSMOS image and caption features under CSA:** The data are inherently noisy, as indicated by the correlation coefficients of the unimodal feature spaces, which concentrate on 0.2 to 0.4. The unimodal encoders here are GTR and DINOv2.

## C    CORRELATION OF FEATURE SPACES

To take a deeper look into unimodal feature spaces, we show the correlation coefficients of COSMOS image and caption features under CSA in Figure 8. The data are inherently noisy, as indicated by the correlation coefficients of the unimodal feature spaces, which concentrate on 0.2 to 0.4. This distribution of correlation coefficients highlights that, despite the fact that the original multimodal data are noisy and show complex correlations, CSA can effectively map them to a multimodal space where the similarity score remains meaningful for the zero-shot downstream tasks.

## D    SENSITIVITY TO HYPERPARAMETER $s$

| Method | $s$ | mAP | Precision@1 | Precision@5 |
|---|---|---|---|---|
| CSA | 10 | 9.5% | 18.1% | 13.4% |
| CSA | 50 | 32.7% | 53.9% | 40.0% |
| CSA | 100 | 36.0% | 58.3% | 42.9% |
| CSA | 200 | 36.6% | 59.3% | 43.4% |
| CSA | 500 | 31.8% | 56.3% | 38.3% |
| CSA | 750 | 27.3% | 50.2% | 33.6% |
| CLIP | ✗ | 73.8% | 92.9% | 77.2% |
| ASIF | ✗ | 14.6% | 25.6% | 20.0% |

(a) Image-to-text retrieval.

| Method | $s$ | Precision@1 |
|---|---|---|
| CSA | 10 | 15.2% |
| CSA | 50 | 41.2% |
| CSA | 100 | 43.8% |
| CSA | 200 | 44.7% |
| CSA | 500 | 41.7% |
| CSA | 750 | 40.1% |
| CLIP | ✗ | 79.5% |
| ASIF | ✗ | 0.1% |

(b) Text-to-image retrieval.

Table 3: **Cross-modal retrieval on Flickr30k under different $s$:** CSA achieves optimal performance at $s = 200$, and its performance degrades with increases and decreases in $s$, illustrating the trade-off characterized in Section 5.

**Retrieval tasks:** We show CSA's sensitivity to the hyperparameter $s$ in terms of the end performance. In Table 3, CSA achieves optimal performance in $s = 200$, and its performance degrades with increases and decreases in $s$, illustrating the trade-off characterized in Section 5. However, for tasks other than retrieval, we find that a larger $s$ improves performance in image classification, mislabeling detection, and misinformation caption detection. This phenomenon is likely due to the trade-off between distinguishability and informative embedding features, namely the distance between features. Although retrieval tasks require a more curated balance between these aspects, other tasks benefit from greater distinguishability of similarity scores.

**Detection tasks:** Table 4 shows the sensitivity to hyperparameter $s$ of CSA on ImageNet mislabeled data and COSMOS misinformative news captions detection. The relation between hyperparameter $s$ and the AUC performance is different from retrieval tasks (Flickr30k and KITTI), as shown in Table 3. In detection tasks, the AUC increases as hyperparameter $s$ increases. We explained this phenomenon in the theoretical analysis with the perspective of feature distinguishability above.

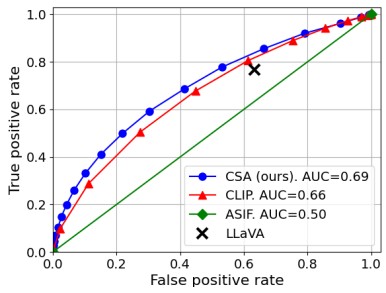 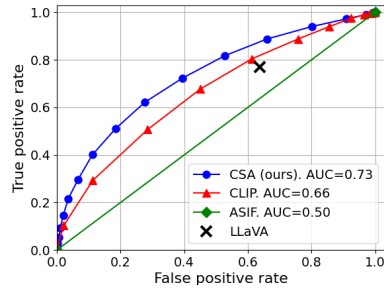

(a) Results for a training size of 5000.      (b) Results for a training size of 35000.

Figure 9: **Detecting mislabeled ImageNet images (cont'd):** CSA (blue) outperforms CLIP, ASIF, and LLaVA with a higher AUC. (a) and (b) illustrate the results for CSA and ASIF across various training set sizes, showing the superior performance of CSA with limited noisy training data. The unimodal encoders are GTR and CLIP (image) here.

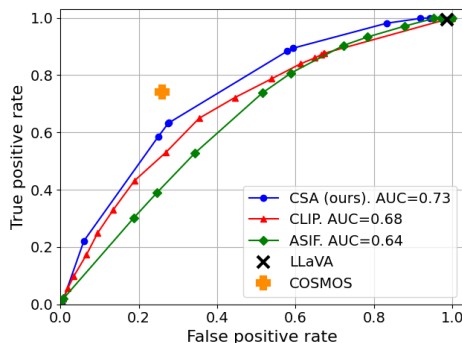

Figure 10: **Detecting misinformative COSMOS captions (cont'd):** CSA (blue) outperforms CLIP, ASIF, and LLaVA with a higher AUC. The supervised-learning method from the original COSMOS paper is the orange cross. It is the only method that outperforms CSA, though trained with supervised labels of object locations. The unimodal encoders are GTR and CLIP (image) here.

## E   SENSITIVITY TO UNIMODAL ENCODERS

We change the unimodal encoders of ASIF and CSA to showcase their generalization ability to different unimodal encoders. Figure 9 shows the results on the detection of mislabeled ImageNet data with other encoders, and Figure 10 shows the results on the detection of misinformative captions with other encoders. CSA again outperforms ASIF and CLIP while outperforming the results of the combination of GTR and DINOv2 in Section 6.

## F   TEXT-TO-LIDAR RETRIEVAL

**Settings:** We additionally test CSA's capability on new, unexplored modality pairs. The KITTI dataset, designed for place recognition in autonomous vehicles, contains RGB and LiDAR images captured in Karlsruhe, Germany. To add a text modality, we use LLaVA Liu et al. (2023) to generate descriptive captions for the images with a prompt "Describe the static objects and the numbers of objects in the image within 20 words." In place recognition, autonomous vehicles utilize local RGB, LiDAR, and text captions to locate themselves within a city by retrieving the nearest landmark from a reference set. The landmarks act as the reference data, while the vehicles' local observations serve as query data, resulting in trimodal data for this task. We used Lip-loc Shubodh et al. (2024), and the image-LiDAR encoder, to encode the LiDAR data and GTR Ni et al. (2022) to encode the captions. The train set contains 5,000 cross-modal instances, and the test set contains 6,000 instances. The

retrieval task here is to locate the vehicle at any landmark within 20 meters. We set this distance threshold per Shubodh et al. (2024).

**Results:** We conducted retrieval on KITTI, augmented with textual captions. Figure 11 shows that CSA achieves the same performance as LiDAR-to-LiDAR retrieval, showcasing CSA's capability of bridging multimodal data. Notably, we are the first to perform text-to-LiDAR retrieval, only made possible by CSA, which maps LiDAR and text embeddings into a shared feature space.

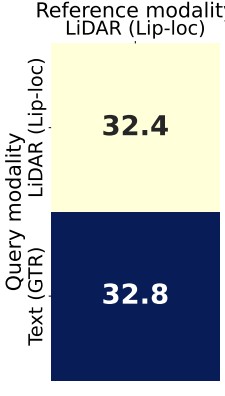

Figure 11: **Text-to-LiDAR retrieval on KITTI:** The figure shows the Recall@5 retrieval rate on KITTI. The cross-modal retrieval result from the LiDAR query is directly from Lip-loc, while the other with text is from CSA. CSA matches the performance of Lip-loc, which is trained on KITTI.

| Method | $s$ | AUC (ImageNet) | AUC (COSMOS) |
|--------|-----|----------------|--------------|
| CLIP | ✗ | 0.66 | 0.68 |
| ASIF | ✗ | 0.50 | 0.64 |
| CSA | 10 | 0.54 | 0.65 |
| | 25 | 0.59 | 0.66 |
| | 50 | 0.62 | 0.68 |
| | 100 | 0.65 | 0.69 |
| | 500 | 0.72 | 0.70 |
| | 700 | **0.73** | **0.70** |

Table 4: **Sensitivity to $s$ on detection tasks:** As analyzed in the appendix of the main paper, the AUC increases as $s$ increases. This trend holds for all detection tasks, which is different from the trend of retrieval.

## G    TEXT-TO-TIMESERIES CLASSIFICATION

**Setting:** We now demonstrate the effect of CSA on more modality pairs. We conducted a classification of handwritten alphabets. One modality is the 3-dimensional time series of movement of pens on a pad Shokoohi-Yekta et al. (2017). The other modality is either the images of alphabets or the text of "Alphabet *X*." We leveraged tsfresh Christ et al. (2018) to extract statistical features from time series. We re-used the same image and text encoder as the main experiments in the paper for the other modalities. Note that tsfresh is not a contrastive-learning encoder but a statistical feature extractor, hence we also demonstrated CSA's ability to adapt any form of encoder.

**Results:** The classification task is similar to that one of ImageNet. We calculated the AUC of multiple classification tasks (an alphabet each) and showed the average AUC under the "ovr (one-vs-rest)" setting. From Table 5, we see that CSA constantly outperforms ASIF by $0.1\times$ of the performance. Notably, we are the first in the community to conduct multimodal classification with multivariate time series, so there are no comparable baselines.

| Method | Modality 1 | Modality 2 | AUC (one-vs-rest) |
|--------|-----------|-----------|-------------------|
| ASIF | time series | image | 0.50 |
| | time series | text | 0.50 |
| CSA | time series | image | 0.58 |
| | time series | text | 0.54 |

Table 5: **Multimodal time series classification:** We further demonstrated CSA's performance on multimodal time series classification, which is first in the community.

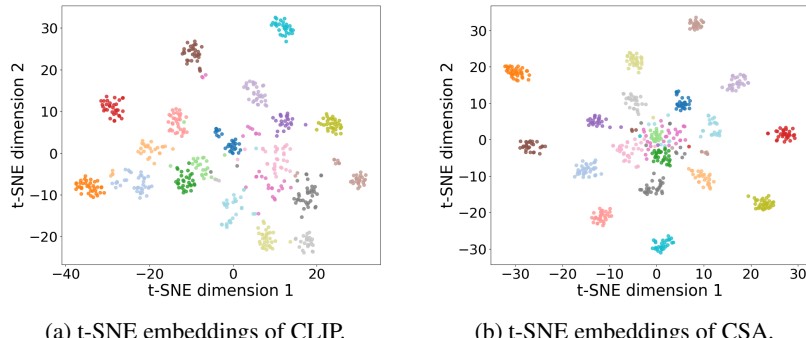

(a) t-SNE embeddings of CLIP.  (b) t-SNE embeddings of CSA.

Figure 12: **t-SNE visualization of embeddings:** (a) shows the t-SNE visualization of CLIP image embeddings, and (b) shows the t-SNE visualization of CSA's embeddings. Both exhibit strong clustering of embeddings, with colors representing their respective classes.

## H  VISUALIZATION OF CSA EMBEDDINGS

In Figure 12, we showed the t-SNE van der Maaten & Hinton (2008) visualization of ImageNet embeddings. We selected 20 classes out of 1000 classes. Both CLIP and CSA have strong clustering of embeddings, thus verifying that they have similar performance on image classification.

## I  LINEAR CLASSIFICATION OF EMBEDDINGS

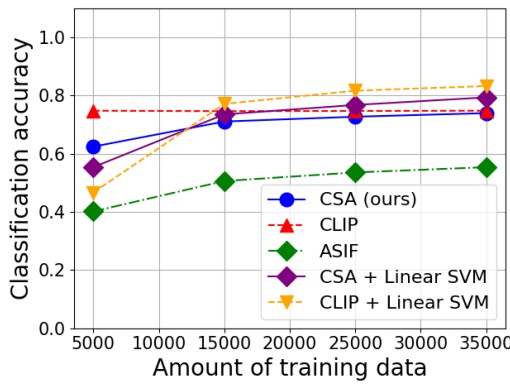

Figure 13: **Classification of SVM with ImageNet embeddings:** Trained linear SVMs enhance classification performance, providing a $10\%$ improvement over zero-shot cross-modal classification. Additionally, CLIP with SVM outperforms CSA with SVM by $5\%$.

We used linear support vector machines (SVMs) to classify CLIP's and CSA's image embeddings. Note that linear classification on image embeddings is fundamentally different from cross-modal classification with text embeddings. This experiment has nothing to do with cross-modal feature space but provides us insight into the clustering of the unimodal embeddings of CLIP and CSA. The results are shown in Figure 13. While CSA outperforms CLIP with SVM when limited to 5k training samples, CLIP with SVM achieves $5\%$ higher classification performance than CSA with SVM when trained on 35k samples, and $10\%$ higher than CLIP without SVM.

## J  ABLATION STUDY ON ENCODER ARCHITECTURES

To further ablate the effect of CSA from the disparity of the unimodal encoder's architecture vs. the multimodal one's. We ran CSA with two different CLIP encoders with the same architecture but different train sets, hence having different feature spaces.

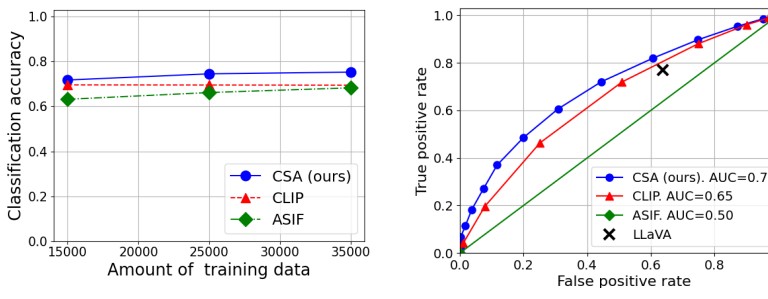

(a) ImageNet classification.  (b) Detecting mislabeled ImageNet images.

Figure 14: **CSA enhances performance across different encoder architectures:** (a) and (b) demonstrate that CSA's success in image classification and mislabeled data detection is due to the method itself, not the architecture of the unimodal encoders.

**Setting:** All encoders are from the OpenCLIP library, and we choose 2 CLIP encoders with the same backbone (ViT-L/14) and trained on DataComp-1B Gadre et al. (2023) and WIT Srinivasan et al. (2021), respectively. The latter is the original model from OpenAI and the text encoder in the ablation study. Note that the models used here are smaller than the ones used in the main paper (ViT-G/14) as there are no multiple ViT-G/14 CLIP models, and the results are not directly comparable. The size of the train set here is 35,000 for detecting mislabeled images.

**Results:** We used the two encoders to re-run the classification and mislabel detection of ImageNet. We show the results in Figure 14. We observe that CSA constantly outperforms CLIP even with the same encoder architecture, which justifies that the success of CSA is due to its method rather than the encoder architectures of the unimodal encoders.

