# OpenReview forum: "CSA: Data-efficient Mapping of Unimodal Features to Multimodal Features"
_ICLR.cc/2025/Conference — ICLR 2025 Poster_

### Official Review · Reviewer_5r8m · 2024-10-29

**Soundness:** 3
**Presentation:** 2
**Contribution:** 3
**Rating:** 6
**Confidence:** 4

**Summary:**

The paper proposes a novel method called Canonical Similarity Analysis (CSA) to improve data efficiency when mapping unimodal features (e.g., from image or text) into multimodal feature spaces, effectively replicating models like CLIP but requiring significantly fewer data using pretrained unimodal models. The method leverages two pretrained unimodal encoders and applies Canonical Correlation Analysis (CCA) to align the features using a small portion of the training set. CSA does not involve training neural networks, only solving a matrix decomposition for feature mapping. The authors demonstrate its effectiveness outperforming a pretrained CLIP in some cases with far fewer multimodal data.

**Strengths:**

- The paper is well-organized, making it easy to follow the proposed approach and experimental setup.
- A key strength of CSA is its remarkable data efficiency, achieving effective mapping and strong results with significantly less training data compared to the baselines.
- Additionally, the method demonstrates robustness in handling noisy and limited multimodal data, which is essential in real-world applications where large correctly labeled datasets are often scarce.
- Finally, the theoretical analysis offers valuable insights into the trade-offs between reducing noise and preserving meaningful similarity scores, further enhancing the paper's contribution.

**Weaknesses:**

- One major issue is that CSA uses different backbones, such as DINOv2-Giant and a GTR-t5-large as unimodal encoders (lines 715-716), compared to the multimodal baseline using a CLIP ViT-bigG/14 pretrained on LAION-2B (line 712). This creates an unfair comparison and makes hard to objectively state the efficacy of the approach.
- Additionally, the paper's performance on cross-modal retrieval tasks, particularly in text-to-image retrieval, is noticeably weaker than CLIP, which limits its impact in this commonly used task.
- Moreover, the greatest weakness is that CSA requires a small portion of training data (i.e. 35k pairs on ImageNet) to solve the matrix decomposition for the features mapping, but all the evaluations are compared to zero-shot CLIP (not fine-tuned on the same portion of training data), which leads to an unfair advantage. Evaluation (e.g Figure 3(a)/3(b) and Table 2(a)/2(b)) should include few-shots adaptation baselines (e.g CoOp [1], CLIP-Adapter [2], etc..) or at least the standard CLIP linear evaluation protocol [3] for a fair comparison with the multimodal baseline. Note that using CSA on 800 pairs of Leafy Spurge could be the main reason for its improved performance against zero-shot CLIP. This is because this data is so different from the training distribution that merely adapting to it leads to the observed improvements.

[1] Learning to Prompt for Vision-Language Models (https://arxiv.org/abs/2109.01134)

[2] CLIP-Adapter: Better Vision-Language Models with Feature Adapters (https://arxiv.org/abs/2110.04544)

[3] Learning Transferable Visual Models From Natural Language Supervision (https://arxiv.org/abs/2103.00020)

**Questions:**

- Q1. Have you considered evaluating CSA using the same backbone as CLIP (e.g. using the same image and text encoders but pre-trained on two different datasets from OpenCLIP)? This would empirically strengthen your hypothesis and demonstrate the effectiveness of your method.
- Q2. Just out of curiosity, how scalable is CSA when handling larger training datasets? How much could it benefit from optimizing the matrix decomposition on a large-scale multimodal dataset?
- Q3. Is it possible to possible to solve CSA on a different dataset from the downstream one? It would be interesting to see whether this improves performance in tasks like cross-modal retrieval, where CSA currently underperforms compared to CLIP.
- Q4. Could the authors better explain why the Leafy Spurge dataset was chosen as an evaluation benchmark, and how its characteristics make it particularly suitable for demonstrating CSA's capabilities?

I would be happy to improve my rating if the authors address my concerns, particularly regarding the fairness of the current comparison.

**Minor observations**:
- Error in Table I. In the supplementary material, it is stated that the CLIP model used is the ViT-bigG-14 trained on LAION-2B. However, in Table I, CLIP is attributed to 12B pairs of training data.
- Error in Table I. Moreover CLIP ViT-bigG-14 is 2.5B of params in total.

---

> ### Author Response · Authors · 2024-11-18
> **Author Response to Reviewer 5r8m**
>
> We express our gratitude to the reviewer for their valuable feedback. We now address each point in detail.
>
>
> **Q1+W1:** In Appendix E, we show the performance of CLIP image encoder + GTR, which partially addresses the architecture issue. In this setting, CSA still outperforms CLIP. Also, the additional experiment on text-LiDAR retrieval. CSA uses embeddings from the LiDAR encoder (Lip-loc) and GTR. It shows performance on par with LiDAR-LiDAR retrieval. In this setting, CSA utilizes the same backbone (the same encoder actually) of the LiDAR encoder.
>
> **W2+W3:** We agree that ASIF is the only fair comparison, and CLIP is unfair. However, fine-tuning CLIP is not entirely fair as well, since it requires GPU and CSA does not. We additionally conducted a fair additional experiment on text-to-LiDAR retrieval (see section A in the global comment), which is the first in the community. Both the LiDAR encoder we used and CSA (text-to-LiDAR) are trained on the same KITTI train set (W3). We show that CSA achieves the same performance as the LiDAR-to-LiDAR retrieval (W2).
>
> Per the suggestion, we used linear support vector machines (SVMs) to classify CLIP’s and CSA’s image embeddings. Note that linear classification on image embeddings is fundamentally different from cross-modal classification with text embeddings. This experiment has nothing to do with cross-modal feature space but provides us insight into the clustering of the unimodal embeddings of CLIP and CSA. The results are shown in section D in the global comment. The results indicate that the SVM trained on CLIP exhibits a 5% higher classification performance compared to CSA when using 35k training samples.
>
> **Q2:** The underlying implementation of CSA is NumPy, which is efficient in handling large matrix operations (in our case SVD) parallelly on multiple CPUs.
> As stated in line 186 in the original draft, the time complexity of the SVD used in CSA is $O(q^1 q^2 r)$, which is roughly the cube of the unimodal encoder output dimension, not the dataset size. For most unimodal encoders, the output dimension is typically around $1000$.
> Also, there are tricks to speed up SVD approximately. For instance, CuPy can dramatically speed up SVD for large matrices using GPUs, and ​​​​randomized SVD algorithms compute approximate results much faster than full SVD. Also, for a given $s$, we only need to obtain the first $s$ singular values, which can also speed up the computation.
> Therefore, SVD is not computationally expensive in CSA.
>
> To further support this, we provide empirical results with the machine described in the appendix. For the largest unimodal encoder size tested (DINOv2 with a dimension of $1536$), the SVD runtime was $0.55$ seconds using CuPy and $0.44$ seconds using NumPy.
> To sum up, we thank the reviewers for pointing this out. We will discuss more about the means to speed up SVD for large matrices in the limitations sections in the next version.
>
> **Q3:** CSA does not have zero-shot capability as it requires training and testing on similar data distributions. Namely, it is not a foundation model, and it bridges multimodal data with unimodal encoders for specific tasks.
>
> **Q4:** Leafy Spurge tests the limits of CSA with extremely limited data. While it is possible that CLIP or DINOv2 were trained on ImageNet, the exact training data for them are not fully revealed. To address this, a key aspect of Leafy Spurge is that it is released after the models we used, allowing for a fair comparison between CSA and CLIP on a dataset on which no unimodal encoder has been specifically trained. Specifically, we aim to isolate the impact of the task from the influence of unimodal encoders and CSA.
>
> ### Minor observations:
> 1. In Table I, we incorrectly attributed CLIP to 12B “samples seen” but not the training size, which should be 2B. Thanks for the correction. We will update the numbers in the next version, but still, CSA requires significantly less multimodal data.
>
> 2. In Table I, we show the CLIP parameters of only one encoder for a fair comparison to other unimodal encoders (half of 2.5B is 1.2B). Thanks for the comment. We will emphasize it in the next version.

---

> ### Comment · Reviewer_5r8m · 2024-11-26
>
> I thank the authors for addressing Q2, Q3, and Q4.
>
> To clarify my earlier question **Q1**: I am suggesting that you consider evaluating your proposed method using both image and text encoders from a CLIP model that are not already aligned (e.g., taking two CLIP models pretrained on different datasets). The aim of this experiment is distinct from what you have already included in your experiments. Comparing the performance with the two original CLIP models, in my view, could further strengthen your work. Please correct me if I am wrong.
>
> I still believe the comparison in Figure 3(a)/3(b) and Table 2(a)/2(b) is unfair and could be misleading.

---

> > ### Author Response · Authors · 2024-11-26
> > **Ablation Study on Encoder Architectures**
> >
> > We thank the reviewer for the clarification.
> >
> > Q1: Yes, we agree, and thanks for the suggestion. Given that the discussion period has been extended, we conducted a fair comparison of CLIP vs. CSA on the same encoder architecture. The results can be found in Appendix E in the updated global comment. We briefly describe the results here.
> >
> > ### Setting:
> > All encoders are from the OpenCLIP library. We chose 2 CLIP encoders with the same backbone (ViT-L/14) and trained on DataComp-1B and WIT, respectively. The latter is the original model from OpenAI. Note that the models used here are smaller than the ones used in the main paper (ViT-G/14) as there are no multiple ViT-G/14 CLIP models, and the results are not directly comparable. The size of the train set here is $35,000$.
> >
> > ### Results:
> > We then used the two encoders to re-run the classification and mislabel detection tasks of ImageNet.
> > We show the results in Appendix E in the revised global comment. CSA shows an AUC of $0.7$ on the mislabeled image detection task while CLIP is $0.65$.
> > On classification, surprisingly, CSA outperforms the CLIP encoder by roughly $5%$, while in the main paper CSA is only on par with CLIP.
> >
> > To sum up, CSA constantly outperforms CLIP even with the same encoder architecture, which justifies the success of CSA is due to its method rather than the encoder architectures of the unimodal encoders.

---

> ### Author Response · Authors · 2024-11-26
> **Addressing the (Un)Fairness**
>
> ### Unfair comparison of Text-image tasks
> As we emphasized in line 363 “*ASIF is the only fair baseline for CSA… We also compared CSA with the state-of-the-art multimodal encoder model from OpenCLIP, acknowledging their incomparable training set and model parameters…*” We have no intention to mislead the readers that CLIP is a fair baseline but emphasize that **CLIP serves as a baseline to highlight the strong performance of CSA** due to ASIF’s significantly lower performance. Also, the use of CLIP on ImageNet and Flickr30k is common in the community, which is also run in the original paper of CLIP and LAION-5B.
>
> Lastly, in the main paper, we compare the size of multimodal training sets between CSA and CLIP to emphasize that CSA can bridge unexplored modality pairs that lack existing multimodal encoders while being performant. This motivation leads to the additional experiments detailed below.
>
> ### Fair Comparison on Training Set, Unfair on Modality: Text-to-LiDAR Retrieval
> Unlike the text-to-image domain, our additional experiments on text-to-LiDAR retrieval (global comment Appendix A) are completely fair in terms of train sets. CSA is on par with multimodal encoders trained on the same dataset. However, the modalities of CSA and Lip-Lock are different, thus unfair in modalities.
>
>
> ### Beyond fairness–Remark on CSA’s contribution
> CSA is a pioneering work that aims to bridge the gap between new modality pairs. Since the problem itself is new, only ASIF is a fair comparison. However, we ambitiously show that CSA can actually achieve the same performance as existing multimodal encoders despite the unfairness in train sets, modalities, and model architectures.
>
> As suggested by the reviewer, the most we can do is conduct fair experiments per factor:
> (1) Appendix A in the global comment ablates the effect of train set.
> (2) Figure (3) and Table (2) in the main paper ablate the effect of modality pairs.
> (3) Appendix E in the global comment ablates the effect of model architectures.
>
> We will clarify more in the next version of the paper and hope our explanation addresses the concerns and questions. We kindly ask the reviewer to reassess their evaluation.

---

> > ### Comment · Reviewer_5r8m · 2024-11-27
> >
> > Thank the authors for providing a detailed response, most of my concerns have been addressed.
> >
> > I will raise my rating in favor of accepting this paper.

---

### Official Review · Reviewer_JqjS · 2024-10-31

**Soundness:** 3
**Presentation:** 3
**Contribution:** 2
**Rating:** 6
**Confidence:** 3

**Summary:**

A new similarity calculation strategy, CSA, based on Canonical Correlation Analysis, is proposed to replace the correlation matrix in CLIP for aligning images and text modalities. CSA allows better alignment of features encoded by unimodal encoders for the same samples without requiring as many training samples as CLIP. A theoretical explanation of CSA is provided. Experiments are conducted on a large number of datasets, all of which outperform the baseline model ASIF.

**Strengths:**

1. It only requires using a pre-trained unimodal encoder along with CSA, training on a small number of sample pairs (such as a single image and its corresponding caption) to achieve alignment between two modalities, greatly saving computational resources.
2. In image classification tasks, it achieves results comparable to CLIP using fewer resources.
3. Experimental results significantly outperform the baseline model ASIF.

**Weaknesses:**

1. The theoretical discussion of CSA lacks detailed explanation for Equation (9), and the connection to contrastive learning is not clearly established, indicating a lack of novelity in CSA.
2. In comparisons with CLIP for image classification tasks, only a limited amount of training data was used, with no comparisons made on larger-scale training datasets.

**Questions:**

See weakness.

---

> ### Author Response · Authors · 2024-11-18
> **Author Response to Reviewer JqjS**
>
> We thank the reviewer for their insightful feedback.
>
> 1. Equation (9) is directly obtained from ​​the ​​previous work cited in line 242.  Contrastive learning essentially enforces two properties: (1) clustering similar data instances and (2) pushing dissimilar instances far away in the feature space. Since the unimodal encoders have both properties, CSA is just re-doing “clustering similar data instances” for multimodal data pairs. CSA does that by mapping the multimodal features to maximize the coefficient correlations, which is partially the objective of contrastive learning. Hence, Appendix E in the main paper shows that CSA works well with various unimodal encoders trained on contrastive learning losses.
>
> 2. We do not consider larger-scale training datasets due to the motivation of CSA–to bridge new and unexplored modality pairs (see our additional experiment on text-LiDAR in the global comment). These modality pairs do not have sufficient data to train a cross-modal encoder. We ran the ImageNet experiment to demonstrate the capability of CSA on the image-text modality pair, and CLIP serves as an unfair baseline to understand more about CSA. In terms of the implementation of CSA on large datasets, the underlying implementation of CSA is NumPy, which is efficient in handling large matrix operations (in our case SVD) parallelly on multiple CPUs.

---

### Official Review · Reviewer_fPRX · 2024-10-31

**Soundness:** 3
**Presentation:** 2
**Contribution:** 2
**Rating:** 6
**Confidence:** 4

**Summary:**

This paper introduces a canonical similarity analysis (CSA) framework, a novel approach that projects two distinct unimodal feature spaces from pre-trained encoders into a unified multimodal feature space. The CSA has great data efficiency and discovers the inherent trade-offs of informative embeddings and distinguishing data. The extensive experiments show that the CSA achieves competitive performances against CLIP models across a range of tasks, including image classification, mislabeled data detection, cross-modality retrieval, and misinformation detection, even with a limited dataset.

**Strengths:**

1. This paper proposes the canonical similarity analysis (CSA) framework, which can replicate the CLIP multi-modal model. It just uses two unimodal encoders but demands much less computation cost and related data.
2. This paper provides the theoretical analysis on the trade-off of obtaining informative embeddings and distinguishing multimodal data, considering various hyperparameters of CSA.
3. The extensive experiments on various downstream tasks (such as image classification, cross-modal retrieval, and misinformative caption detection) show that CSA outperforms the traditional CLIP model, while requiring much fewer paired multimodal data and fewer unimodal data.

**Weaknesses:**

1. This paper proposes a post-tuning mapping framework on the unimodal features, which would compute the matrix optimization without training any encoders. Hence the performances heavily rely on the choices of visual and textual encoders. The experiments part only shows the one encoder situation (gtr + dino). More model encoders analysis for CSA are needed.
2. In the performance comparisons, the ASIF is the only fair baseline method, which lacks of persuasion. It is important to add more comparative experiments for more related methods, e.g., some prompt tuning and adapter tuning work (PEFT methods). What’s more, when compare CSA with CLIP (e.g., Flickr in Tab. 2, ImageNet in Fig. 3), the CSA is the fine-tuned model on the specific dataset, but the CLIP is the zero-shot model without fine-tuning. Such experimental comparison is unreasonable. The CSA should fine-tuning on the ImageNet and then test on the Flickr.
3. The feature dimensions of $q$ and $r$ are the important hyper-parameters for CSA, and the Tab.1 also shows the different dimension choices on different datasets, hence the related ablation study is missing in this paper.

**Questions:**

1. The visualization experiments can show the interpretability of multi-modal learning, I wonder to know the visualization results of embedding spaces.
2. Although the CSA does not require GPUs to optimize the fitting matrix, I wonder to know the run-time cost of the optimization process on different datasets and feature dimensions. Is it faster than the training a traditional network?

---

> ### Author Response · Authors · 2024-11-18
> **Author Response to Reviewer fPRX**
>
> We thank the reviewer for their insightful feedback.
>
>
> W1: As mentioned in line 473 in the main paper, we conducted an ablation study on various unimodal encoders in Appendix E. Also, our additional experiment (see section A in the global comment) on text-to-LiDAR retrieval uses a LiDAR-image encoder, Liploc, and GTR for text. It also shows the generalization ability for various unimodal encoders. As discussed in the main paper, contrastive learning-based encoders should all work smoothly with CSA, and our experimental results support this point.
>
> W2: We agree that ASIF is the only fair comparison. We highlight that fine-tuning CLIP is not entirely fair as well, since it requires GPU and CSA does not. Also, we do not think CSA has any zero-shot capability as it requires training and testing on similar data distributions. Hence, it will likely not perform well when fine-tuned on ImageNet and then tested on Flickr. However, we conducted a fair additional experiment on text-to-LiDAR retrieval, which is the first in the community (see section A in the global comment). Both the LiDAR encoder used and CSA (text-to-LiDAR) are trained on the same train set of KITTI. This setup emphasizes the point of training the cross-modal encoder and CSA on the same dataset. We show that CSA achieves the same performance as the LiDAR-LiDAR retrieval.
>
> W3: $q$ is not a hyper-parameter. It represents the fixed output dimensions on the unimodal encoders. $r$ is the output dimension of CCA, which does not affect the whole CSA pipeline. The only hyperparameter is $s$, which we conducted the ablation study in Appendix D. $r$ does not affect the overall results because only $s$ dimensions from the $r$ dimensional outputs are used to calculate the similarity in Eq. 4.
>
> Q1: In Fig. 2 and section C in the global comment, we showed the t-SNE visualization of ImageNet embeddings with CLIP and CSA. We selected $20$ classes out of $1000$ classes. Both CLIP and CSA have strong clustering of embeddings, thus verifying that they have similar performance on classification.
>
> Q2: Yes, as indicated in line 710 in the main paper, solving Equation 2 with $35,000$ multimodal feature vectors on a 64-core Xeon Gold 6226R CPU machine takes less than 10 minutes, which is significantly faster than any GPU training. Per the request of other reviewers, we also numerically showed how long it takes to solve CSA on ImageNet here: https://openreview.net/forum?id=6Mg7pjG7Sw&noteId=Siu0an4HUF

---

### Official Review · Reviewer_Zosg · 2024-11-02

**Soundness:** 3
**Presentation:** 4
**Contribution:** 3
**Rating:** 6
**Confidence:** 3

**Summary:**

The paper proposes CSA: Canonical Similarity Analysis method to train a multimodal encoder from two independent unimodal encoders with limited data. Unlike traditional methods that train both encoders on multimodal data, CSA only requires the unimodal encoders to generate embeddings, avoiding the need to train them on multimodal pairs.

The authors show that their CSA method matches or beats CLIP and the baseline ASIF on image classification on ImageNet and LeafySpurge datasets. However, the authors’ method falls short of CLIP’s performance on the cross-modal retrieval task. Their experiments also show that CSA outperforms the baselines when detecting mislabeled ImageNet images, and detecting misinformative captions.

The paper concludes that CSA is a robust method to replicate CLIP similarity scores using two pre-trained unimodal encoders and limited data.

**Strengths:**

- The paper is well motivated: Finding data-efficient ways to train multimodal models using existing pretrained unimodal encoders is an important research direction.
- The method used in the paper appears novel, with only one other related work (ASIF) that uses independent unimodal encoders to project embeddings onto a multimodal representational space.
- The proposed approach (CSA) beats or matches CLIP’s performance in image classification tasks using significantly lesser data.
- The paper’s experiments also show that CSA is more robust to mislabeled images in the image classification task. CSA is also slightly more capable in detecting misinformative captions compared to CLIP and other baselines. Finally, the authors show that CSA is more robust to noisy data compared to the previous baseline ASIF.
- Overall, this study could pave the way for more advancements in this space of adapting independent unimodal encoders to multimodal models.

**Weaknesses:**

- In Figure 3, CSA was only trained on in-distribution image-caption pairs. This may lead to an unfair comparison to CLIP, as the CSA training has seen the ImageNet/Leafy Spurge distribution that its being tested on during the training process. CLIP’s rise to fame is due to its general zero-shot capabilities. The zero-shot capabilities of CSA are not fully evaluated in this paper.
    - A fairer comparison might be to fine-tune CLIP on ImageNet/Leafy Spurge. This could be done by training the last projection layer of CLIP, analogous to keeping the unimodal encoders frozen in CSA training.
    - Additionally, including out-of-distribution datasets would give a clearer view of CSAs true zero-shot capabilities: An example could be to train CSA on MS COCO and then evaluate the performance on ImageNet / Leafy Spurge.

- Similarly, in Figure 4, comparing CLIP to CSA may not be entirely fair. Although CLIP likely encountered ImageNet images in training, they represent a very small fraction of its large dataset. A fairer comparison could be to fine-tune CLIP (again, perhaps just the  projection layer) on ImageNet images, including those with mislabeled data, before proceeding with this analysis (See Question 2).

- The paper's conclusion suggests that CSA outperforms CLIP in cross-modal retrieval ("CSA shows competitive performance compared to CLIP models in image classification, mislabeled data detection, **cross-modality retrieval**, and misinformation detection tasks with
limited data."). However, this is misleading, as Table 2 shows that CSA underperforms CLIP in this task.

**Questions:**

1. In Figure 4, does the smaller subset of ImageNet (5k images) include all of the mislabeled images from ImageNet? A statistic showing the percentage of mislabeled images in the training set for Figures 4(a) and 4(b) would offer more insight into these results.
2. Why did the authors choose not to fine-tune CLIP on ImageNet? Was there some intuition behind this?

---

> ### Author Response · Authors · 2024-11-18
> **Author Response Reviewer Zosg**
>
> We thank the reviewers for their detailed comments.
>
> ## Weakness:
>
> 1. CSA bridges multimodal data with unimodal encoders. We do not think CSA has any zero-shot capability like foundation cross-modal models. Just like most machine-learning models, it trains (solving CCA) and tests on similar data distributions. Hence, the way to use CSA is to train it with limited data of new modality pairs.
>
> 2. We highlight that the only fair comparison is ASIF, as CLIP can never be entirely fair, even with fine-tuning, which requires GPU. However, we conducted a fair experiment on text-to-LiDAR retrieval, which is the first in the community (section A in the global comment). The LiDAR encoder used and CSA (text-to-LiDAR) are trained on the same subset of KITTI. We show that CSA achieves the same performance as the LiDAR-image encoder on the retrieval task.
>
> 3. That is correct, but our additional result shows that CSA matches the performance of cross-modal encoders in text-to-LiDAR retrieval (see section A in the global comment). We will address it in the next version of the paper to clarify.
>
> ## Questions:
> 1. No, the 5k images are also randomly sampled from the whole dataset, which means that the percentage of mislabeled images (10.9%) is the same as in Fig. 4(b). Since empirically, one does not know the mislabeled data a priori, thus we think it is more reasonable to show that the trainsets contain a fixed percentage of mislabeled data. We will address them in the next version of the paper.
>
> 2. We did not fine-tune CLIP for two reasons: (1) The essence of CSA is to bridge multimodal data with unimodal encoders without GPU training, so we do not think it is fair to fine-tune CLIP as it requires GPU. (2) We envision that CSA can be used to bridge unexplored modalities, where there are no cross-modal encoders like CLIP. The comparison to CLIP is to demonstrate that CSA is performing reasonably well. The additional text-to-LiDAR retrieval experiment emphasizes this point as there are no text-to-LiDAR encoders designed for retrieval now.

---

> > ### Comment · Reviewer_Zosg · 2024-11-27
> > **Response to authors**
> >
> > I thank the authors for their response.
> > My questions have been sufficiently answered.
> >
> > In terms of addressing weaknesses
> >
> > W1 and W2: I understand the authors' reasoning, suggesting that CSA does not have any zero-shot abilities as CLIP. However, the purpose of comparing with fine-tuned CLIP was to provide a fairer comparison between the 2 methods when they have been trained on the same dataset. The text-to-LiDAR evaluations address this concern, because the LiDAR encoder and CSA (text-to-LiDAR) are trained on the same subset of KITTI..
> >
> > W3: While the additional results on text-to-LiDAR retrieval help strengthen CSA's performance claims, the poor cross-modal retrieval performance compared to CLIP is still concerning, especially considering that CSA was trained on the Flickr30k dataset, whereas CLIP is being tested in zero-shot manner. Some discussion pertaining to the discrepancy in performance between cross-modal retrieval (CSA performs worse than CLIP zero-shot) and image classification (CSA performs better than CLIP zero-shot) would be helpful in improving the analysis.
> >
> > I will maintain my positive score for now.

---

> > > ### Author Response · Authors · 2024-11-27
> > > **Discrepancy in Performance between Retrieval and Classification**
> > >
> > > We thank the reviewer for the response.
> > >
> > > W3: Yes, we believe the discrepancy in performance between cross-modal retrieval and image classification needs more investigation, as the text-to-LiDAR retrieval task performs on par with multimodal encoders. However, we conducted an analysis in Appendix D in the original submission, which explained the trade-off between distinguishability and informative embedding features, as motivated in the theoretical analysis section.
> > >
> > > We kindly request the reviewer to refer to Appendix D and share any additional concerns or feedback.

---

> > > > ### Comment · Reviewer_Zosg · 2024-12-02
> > > >
> > > > Thank you for the response. The discussion in Appendix D is helpful. Most of my concerns have been addressed. I will raise my score.

---

### Official Review · Reviewer_Wv4t · 2024-11-02

**Soundness:** 3
**Presentation:** 3
**Contribution:** 2
**Rating:** 6
**Confidence:** 3

**Summary:**

The paper proposes a new approach called Canonical Similarity Analysis (CSA) that addresses the challenge of training multimodal encoders, like CLIP, which typically require large amounts of paired multimodal data. CSA achieves efficient multimodal mapping by using two unimodal encoders, thereby reducing data requirements significantly. CSA utilizes canonical correlation analysis to map unimodal features into a multimodal space and introduces a weighted cosine similarity function to replicate multimodal similarity scores. Experiments across tasks such as image classification, misinformation detection, and cross-modal retrieval demonstrate CSA’s data efficiency and robustness, especially in scenarios with limited or noisy data.

**Strengths:**

S1: (**Data Efficiency and Good Performance**) CSA requires significantly fewer multimodal data pairs by relying on unimodal encoders, which could benefit researchers constrained by data or computational resources. It needs as little as 35,000 image-text pairs to match the performance of CLIP on ImageNet, which is especially notable.

S2: (**Computational Simplicity**) One of the most accessible aspects of CSA is its ability to function effectively without requiring GPU-intensive training, which is a significant advantage for researchers with limited computational resources.

S3: (**Theoretical Analysis**)The authors include a theoretical analysis for CSA using canonical correlation analysis.

**Weaknesses:**

W1: (**Hyperparameter Sensitivity and Limited Justification for $s$ Selection**)
The choice of the hyperparameter $s$ in the canonical similarity metric (Section 4.2) lacks comprehensive justification. While the authors discuss a trade-off in feature distinguishability based on $s$, a more detailed sensitivity analysis showing how varying $s$ affects downstream performance across tasks would add clarity, as the framework is quite sensitive to $s$ indicated by content in Table 3. Table 3 provides some insights, but a broader, task-specific evaluation could make the effects of this parameter choice clearer and improve reproducibility for other researchers.


W2: (**Weakness on Number of Modalities Supported**) While CSA shows promising results with image-text pairs and briefly explores audio-text applications in the appendix, the experimental evaluation of additional modalities remains limited and lacks depth. The audio experiments, while mentioned, do not provide a strong enough demonstration of CSA’s effectiveness beyond the main image-text modality. This limited exploration weakens CSA’s claim of generalizability across modalities and would benefit from more robust testing on diverse modality pairs to strengthen its applicability in multimodal contexts.


W3: (**Scalability Concerns**) While CSA requires fewer training samples to match performance on certain benchmarks (e.g., ImageNet classification with 35,000 image-text pairs), some datasets or applications may still demand larger sample sizes to achieve comparable performance, especially those with more complex or diverse data structures. For instance, datasets like OpenImages (available at https://storage.googleapis.com/openimages/web/index.html), which contain extensive variety in both images and captions and a wide array of object categories, may require larger data samples for CSA to perform effectively. In such cases, the scalability limitations of CSA's matrix decomposition approach become more evident, as the cubic complexity of these operations could lead to significant computational strain on larger datasets. More experimental validation on these varied datasets would clarify the scalability and generalizability of CSA across a broader range of applications.

**Questions:**

The reviewer will consider increasing the score if the authors can address the weaknesses mentioned above, most likely empirically.

---

> ### Author Response · Authors · 2024-11-18
> **Author Response to Reviewer Wv4t**
>
> We thank the reviewer for appreciating our paper. To address the weakness:
>
>
> W1: We added an additional comparison on hyperparameter $s$ vs. the AUC of detection tasks, as discussed in Section B of the global comment. The results align with the statement in the original paper: larger hyperparameter $s$ that matches the dimension of unimodal encoders is better for detection tasks.
> Whereas in the cross-modal retrieval task on Flickr30k, $s$ should be in the middle between the lowest and highest dimensions ($0 < 200 < 768$=GTR’s dimension). We also observe a similar trend in the additional experiment on text-to-LiDAR retrieval with the KITTI dataset (section A in the global comment). Again, we select a medium $s=100$, where the maximum dimension of the LiDAR encoder is $256$.
>
>
> W2: Yes, we agree with the reviewer. We additionally tested CSA’s capability on new, unexplored modality pairs. We conducted text-to-LiDAR retrieval on an autonomous driving place recognition dataset (section A in the global comment). We showed that CSA achieves the same performance as LiDAR-to-LiDAR retrieval, showcasing CSA’s capability. Notably, we are the first to perform text-to-LiDAR retrieval, only made possible by CSA, which maps LiDAR and text embeddings into a shared feature space.
>
>
> W3: Due to time constraints, we cannot run the OpenImages dataset with CSA. However, we emphasize that the underlying implementation of CSA is NumPy, which is efficient in handling large matrix operations (in our case SVD) parallelly on multiple CPUs. Using NumPy to run SVD is far more energy, time, and computationally efficient than training deep learning models on GPUs for datasets of the same size. Last but not least, there are tricks to speed up SVD. For instance, CuPy can dramatically speed up SVD for large matrices using GPUs, and ​​​​randomized SVD algorithms compute approximate results much faster than full SVD. Also, for a given s, we only need to obtain the first s singular values, which can also speed up the computation.
> Lastly, CSA aims to bridge unexplored modality pairs, which are naturally insufficient in data. In such cases, one does not need to worry about the computation time and the scalability of CSA.
> We thank the reviewers for pointing this out. We will discuss more about the means to speed up SVD for large matrices in the limitations sections in the next version.

---

> > ### Comment · Reviewer_Wv4t · 2024-11-24
> > **Response to Authors**
> >
> > Thanks for the detailed comments on my concerns!
> >
> > For W1: my concern is generally addressed.
> >
> > For W2: I think this is still a limitation of this work, given the good result on the text-to-LiDAR task included.
> >
> > For W3: I am not fully convinced since the approximate algorithms also bring errors. It would be better to see the empirical results of the proposed method in this case. However, the reviewer understands it might not be feasible during the rebuttal period.
> >
> > In sum, I will keep my positive score here.

---

> ### Author Response · Authors · 2024-11-25
> **Clarification on CSA's time complexity and scalabiltiy**
>
> We thank the reviewer for highlighting concerns about the time complexity of CSA. To address W3, we would like to clarify a potential misunderstanding.
>
> ## Emerging Modality Pairs vs. Scalability
> One important usage of CSA is to bridge new modality gaps, such as LiDAR and text.
> In these situations, the lack of multimodal data does not pose scalability challenges.
>
> ## Time Complexity
> Even when multimodal data is abundant, CSA can still operate efficiently.
> As stated in line 186 in the original draft, the time complexity of the SVD used in CSA is $O(q^1 q^2 r)$, roughly the cube of the unimodal encoder output dimension, not the dataset size. For most unimodal encoders, the output dimension is typically around $1000$. Therefore, SVD is not computationally expensive in CSA.
>
> To further support this, we provide empirical results with the machine described in the appendix. For the largest unimodal encoder size tested (DINOv2 with a dimension of $1536$), the SVD runtime was $0.51$ seconds using CuPy and $0.44$ seconds using NumPy.
>
> Hence, the scalability of CSA (or essentially SVD) should not be a major concern.

---

> ### Author Response · Authors · 2024-11-25
> **Expanding Modality Bridging: CSA's Capabilities and Innovations**
>
> To address W2, we acknowledge that CSA is just a first step towards bridging new modality pairs, and we expect more work in the future to tackle it. However, as demonstrated in our additional experiments, we are the first to perform text-to-LiDAR retrieval, a task that previous works cannot achieve.
>
> Regarding the number of modalities supported by CSA, it handles any modality as long as there are unimodal encoders trained with a contrastive loss for that modality. In such cases, CSA should be able to connect it with other modalities effectively.
>
> We kindly ask the reviewer to reassess their evaluation if we have addressed the weaknesses mentioned.

---

> > ### Comment · Reviewer_Wv4t · 2024-11-27
> >
> > Thanks for the reply.
> >
> > For the complexity, I agree it is not a major concern based on the authors' reply.
> >
> > But for more modalities, though the reviewer agrees with the authors on the contribution of the text-to-LiDAR task, but still, the major claim of the paper is to connect "multimodal features", there are definitely more modalities than just image and LiDAR (and unimodal encoders available). Missing empirical results on those hurt the contribution of this work a bit, as "can be generalized to different modalities" is not 100% equal to "work well on them".

---

> ### Author Response · Authors · 2024-11-28
> **More Modalities--Time Series**
>
> **Setting:**
> We now demonstrate the effect of CSA on more modality pairs, as suggested by the reviewer.
> We conducted a classification of handwritten alphabets. One modality is the $3$-dimensional time series of movement of pens on a pad [2]. The other modality is either the images of alphabets or the text of "Alphabet *X*."
> We leveraged tsfresh [1] to extract statistical features from time series and re-used the same image and text encoder as the main experiments in the paper for the other two modalities.
> Note that tsfresh is **not a contrastive-learning encoder** but a statistical feature extractor, hence we also demonstrated CSA's ability to adapt **any general form of encoders (feature extractors)**.
>
> **Results:**
> The classification task is similar to the one of Imagenet. We calculated the AUC of multiple classification tasks (an alphabet each) and showed the average AUC under the "ovr (one-vs-rest)" setting.
> From Table 2 in the global comment, we see that CSA constantly outperforms ASIF by $4$~$8$% in AUC. Notably, we are the first in the community to conduct multimodal classification with multivariate time series, so there are no comparable baselines.
>
> **Remarks on modalities:**
> We have showcased CSA's versatility across various modalities, including (1) images, (2) text, (3) audio, (4) time-series data representing 3-dimensional hand movements, and (5) LiDAR. Furthermore, CSA can be applied to other general time-series modalities, such as audio, IMU data, human body motion, and even object outlines in images [3].
>
> We believe this comprehensive demonstration addresses the reviewer’s concerns regarding the range of modalities supported by CSA. We kindly request the reviewer to reconsider their evaluation since we have addressed all the concerns.
>
> Reference:
>
> [1] Christ, Maximilian, et al. "Time series feature extraction on basis of scalable hypothesis tests (tsfresh–a python package)." Neurocomputing 307 (2018): 72-77.
>
> [2] Shokoohi-Yekta, Mohammad, et al. "Generalizing DTW to the multi-dimensional case requires an adaptive approach." Data mining and knowledge discovery 31 (2017): 1-31.
>
> [3] Middlehurst, M. and Schäfer, P. and Bagnall, A. (2024). Bake off redux: a review and experimental evaluation of recent time series classification algorithms. Data Mining and Knowledge Discovery, online first, open access.

---

> > ### Author Response · Authors · 2024-12-02
> >
> > Dear Reviewer Wv4t,
> >
> > We wanted to kindly follow up regarding your review of our submission. We have provided additional results with more modalities, per your suggestion. Please let us know if you need any clarification from our side.
> >
> > We kindly request the reviewer to reconsider their evaluation if we have addressed all the concerns.
> >
> > Authors

---

### Author Response · Authors · 2024-11-18
**Additional Experiments**

Please refer to the link below for results on additional experimental results. We also uploaded a revised submission per the reviewers' feedback and highlighted the changes in red.

* Note on 11/26/2024: Updated results on CLIP with the same backbone architecture (Sec. E).
* Note on 11/28/2024: Updated results on multimodal time series classification (Sec. F).

https://drive.google.com/file/d/1KlhmpckDFuFkvobmA6EBEWl91Mc7X_G7/view?usp=sharing

---

### Meta-Review · Area_Chair_8VPe · 2024-12-16

**Metareview:**

This paper aims to design an efficient scheme for fast multi-modal learning. With frozen CLIP or the like, they build their model with canonical similarity (deduced from CCA) to measure the similarity from unimodal to multimodal feature spaces. As the authors claimed, the main strength of this paper lies in using fewer multi-modal data pairs to achieve SOTA downstream task performance.

All the reviewers have main concerns about the extension capabilities of using more modalities or limited kinds of data pairs, as well as the parameter sensitivity of $s$, which is the key hyperparameter for this work. All the reviewers pointed out the advantages of this paper.

To me, if the authors still highlight the $50,000$ times fewer multimodal data pairs, compared to the CLIP, I think it may be misconducted because the proposed method used the pre-trained unimodal encoders, rather than raw features, which is different from training from raw. Therefore, it is suggested that these overexaggerated words be removed, rather than highlighted, in the final version.

Generally, based on the comments and the authors' rebuttal, this is good work to be accepted for ICLR 2025.

**Additional Comments On Reviewer Discussion:**

All the reviewers raised some concerns about the technical and experimental parts of this paper, and the authors responded to all these comments point to point. Although some reviewers would like to raise their scores, all these actions are not shown in the final sheet. As the authors pointed out, the reviewers said they would promote their evaluations yet no clear actions, maybe the timeline or other limitations. No matter what happened, the overall idea of this paper is easy to understand and has good performance on two different tasks, zsl and cross-modal retrieval, with fewer data pairs. This work deserves to be presented on ICLR.

---

### Decision · Program_Chairs · 2025-01-22

Accept (Poster)